Comparative chemical analysis of army ant mandibular gland volatiles (Formicidae: Dorylinae)

http://orcid.org/0000-0002-9184-8562 Brückner Adrian 1 2 adrian.brueckner@gmail.com
Hoenle Philipp O. 1
von Beeren Christoph 1 cvonbeeren@gmail.com
1 Ecological Networks, Technische Universität Darmstadt , Darmstadt , Germany
2 Division of Biology and Biological Engineering, California Institute of Technology , Pasadena, CA , USA
Pie Marcio
Electronic publication date: 2018 Jul 16
Publication date: 2018
Volume: 6
Electronic Location ID: e5319
Received 2018 May 22; Accepted 2018 Jul 3
Copyright: © 2018 Brückner et al.
Copyright year: 2018
Copyright holder: Brückner et al.
License: This is an open access article distributed under the terms of the Creative Commons Attribution License, which permits unrestricted use, distribution, reproduction and adaptation in any medium and for any purpose provided that it is properly attributed. For attribution, the original author(s), title, publication source (PeerJ) and either DOI or URL of the article must be cited.
License URL: https://creativecommons.org/licenses/by/4.0/

Keywords: Neotropics, Biochemical systematics, Chemical ecology, Ecitonini, Volatile organic compounds, Pheromones, Eciton burchellii, Alarm response, Chemical communication

Funding: German National Academic Foundation German Research Foundation (Deutsche Forschungsgemeinschaft; No. BE 5177/1-1 and BE 5177/3-1) German Research Foundation Open Access Publishing Fund of Technische Universität Darmstadt Adrian Brückner was supported by the German National Academic Foundation (Studienstiftung des deutschen Volkes) and Christoph von Beeren by the German Research Foundation (Deutsche Forschungsgemeinschaft; No. BE 5177/1-1 and BE 5177/3-1). This work was also supported by the German Research Foundation and the Open Access Publishing Fund of Technische Universität Darmstadt. The funders had no role in study design, data collection and analysis, decision to publish, or preparation of the manuscript.

==============================
Army ants are keystone species in many tropical ecosystems. Yet, little is known about the chemical compounds involved in army ant communication. In the present study, we analyzed the volatile mandibular gland secretions—triggers of ant alarm responses—of six Neotropical army ant species of the genus Eciton (outgroup: Nomamyrmex esenbeckii). Using solid-phase microextraction, we identified 12 chemical compounds, primarily ketones with associated alcohols, one ester and skatole. Most compounds were shared among species, but their relative composition was significantly different. By comparing chemical distances of mandibular gland secretions to species divergence times, we showed that the secretions’ compositions are not strictly determined by phylogeny. By identifying chemical bouquets of seven army ant species, our study provides a valuable comparative resource for future studies aiming to unveil the chemicals’ precise role in army ant alarm communication.

Introduction

Exocrine gland secretions play fundamental roles in interactions of arthropods with each other and with their environment (Eisner, 2003). Plenty of functions are known for these secretions including venoms of scorpions or sugary exudates in ant-associated caterpillars (Eisner, 2003). Particularly common are defensive secretions, which often are evolutionarily conserved (Hefetz, 1993; but see Brand, 1978). Besides their role in repelling opponents, defensive secretions in social insects often serve as alert signals to nestmates (e.g., formic acid in formicine ants; Wilson & Regnier, 1971). In contrast to these unspecific alarm-defense compounds, many social insects additionally evolved specific alarm signals aiming to recruit nestmates (“aggressive alarm” sensu Wilson & Regnier, 1971). For instance, “aggressive alarm” pheromone blends of fungus-growing ants were mostly species-specific (Norman et al., 2017). Supposedly, natural selection drives such aggressive alarm signals to diversify among sympatric ant species (see Leonhardt et al., 2016; Wilson & Regnier, 1971).

In ants, mandibular gland secretions generally contain alarm pheromones (Hölldobler & Wilson, 1990; Wilson & Regnier, 1971). In Eciton army ants, Brown (1959) observed that crushed heads of Eciton hamatum majors stimulate an “aggressive alarm” response resulting in massive recruitment of workers and vigorous attacks, bites, and stings—a phenomenon we similarly observed with various Eciton species (see also Torgerson & Akre, 1970). In contrast, headless bodies evoked little or no response by passing army ants (Brown, 1959). Later, a study by Torgerson & Akre (1970) demonstrated that workers usually missed to show a typical alarm response when confronted with crushed major heads of allospecific army ant species, indicating a certain level of pheromone specificity. Furthermore, a study by Lalor & Hughes (2011) indicated that 4-methylheptan-3-one might play an important role for the alarm behavior in Eciton burchellii and E. hamatum. However, the response of workers towards comparatively large amounts (10 μl) of the pure, synthetic ketone was weaker compared to the workers’ response towards crushed sub-major heads (Lalor & Hughes, 2011). We re-tested behavioral responses of six Eciton species (including E. burchellii and E. hamatum workers) in laboratory nests at our field site in Costa Rica towards high doses of 4-methylheptan-3-one (kindly provided by Stefan Schulz, TU Braunschweig), which, however, elicited no pronounced response in workers. In contrast, intracolonial trials with crushed major heads elicited strong excitement among colony members and led to recruitment of major workers. These initial trials led us to re-evaluate the mandibular gland secretions of Eciton army ants.

Additionally, we asked whether the ants’ phylogenetic relationships determine their mandibular gland chemistry. There are two principal ways in which pheromone blends can evolve (Symonds & Elgar, 2008): (1) little and gradual differences in pheromone blends result in phylogenetic clustering (sensu Ivens et al., 2016), that is, closely related species are also most similar in their pheromone blends; (2) major “saltational” shifts in pheromone composition results in phylogenetic overdispersion (sensu Ivens et al., 2016), that is, distantly related species are more similar in their pheromonal blends than closely related species. To assess whether alarm pheromones in army ants rather follow the first (gradual shifts) or the second mode (saltational shifts) of pheromone evolution, we analyzed the volatile mandibular gland chemistry of six sympatric species of Eciton army ants and one species of Nomamyrmex army ant.

Materials and Methods

Ant collection

Most Eciton army ants possess distinct morphological worker castes including large-bodied majors (Powell & Franks, 2006). With their sharply-pointed, sickle-shaped mandibles, the main task of majors is to defend the colony (Fig. 1A). Majors are known to emit alarm pheromones from their mandibular glands when disturbed and therefore our study focused on this particular caste (Brown, 1959). We collected majors of the army ant species Eciton burchellii foreli Mayr 1886, Eciton dulcium crassinode Borgmeier 1955, Eciton hamatum Fabricius 1781, Eciton lucanoides conquistador Weber 1949, Eciton mexicanum s. str. Roger 1863, and Eciton vagans angustatum Roger 1863. Collections took place between 8:00 pm and 3:00 am in the tropical rainforest at La Selva Biological Station, Costa Rica (N10°25.847′ W84°00.404′, altitude 67 m asl) in an area of 11 km2 from February to April 2017. Voucher ant specimens are stored in absolute ethanol and deposited in CvB’s personal collection.

Figure 1 Eciton army ants and their mandibular gland compounds.

(A) Eciton lucanoides major with sickle-shaped mandibles guarding the colony’s emigration column (La Selva, Costa Rica). Photograph by Philipp O. Hoenle. (B) NMDS ordination plot depicting the distinct composition of mandibular gland profiles of different army ant species. Chemical compounds that significantly contributed to data separation are mapped onto the ordination as vectors; compound IDs correspond to Table 1.

We aimed to collect majors from various colonies of a given species. However, ensuring that army ant collections derive from different colonies is notoriously difficult when sampling over a period of several months in a restricted area due to the ants’ migratory habit. Hence, re-sampling of the same colony might have occurred. Due to the high army ant population density at La Selva (O’Donnell et al., 2007), we presume that the majority of colonies were only sampled once. Overall, we collected majors from four collection events for E. burchellii, two collection events for E. dulcium, two collection events for E. lucanoides, four collection events for E. hamatum, four collection events for E. vagans, and one collection event for E. mexicanum (for a map of collection sites and collection dates see Supplemental Information S2). The army ant Nomamyrmex esenbeckii wilsoni Santschi 1920 (one collection event) served as outgroup. This species does not possess a distinct major caste and we thus collected the largest ant workers encountered during a single raid. Specimens were directly frozen and stored at −20 °C and shipped to Germany on dry ice. Ants were identified using the identification key of Longino (2010). Research and export permits were issued and approved by the Ministry of the Environment, Energy and Technology of the Republic of Costa Rica (MINAET; permit numbers: 192-2012-SINAC and R-009-2014-OT-CONAGEBIO).

Chemical analyses of pheromone secretions

Specimens for chemical analysis of mandibular gland contents were haphazardly chosen from the above-mentioned collections by taking majors from as many collection events as possible (see Supplement Information S2). For each head space analysis, we used one ant head except for the smaller species E. mexicanum and N. esenbeckii where two heads were used. We analyzed five head space profiles per species in total except for E. mexicanum (four head space samples corresponding to eight major heads). Heads were removed from the rest of the body, crushed with forceps and placed in a four milliliter glass vial which was immediately sealed with Parafilm® (Bemis, Neenah, WI, USA). Head crushing is a commonly used technique to liberate the volatile organic compounds (VOCs) produced in ants’ mandibular glands and no chemical differences between crushed heads and dissected glands have been reported in Eciton and other ants (Do Nascimento et al., 1993; Hughes, Howse & Goulson, 2001; Keegans et al., 1993). We collected the gland’s VOCs over a period of 30 min from the headspace of crushed heads using a solid-phase microextraction (SPME) fused silica fiber coated with 65 μm polydimethylsiloxane/divinylbenzene (Supelco®; Sigma-Aldrich, St. Louis, MO, USA). Additionally, three air-blank controls were sampled with the same method to ensure that the detected compounds did not derive from any laboratory contamination. Chemical analyses were performed on a QP 2010ultra GC-MS (Shimadzu, Kyōto, Japan). For substance desorption we placed the SPME fiber in the injector port (250 °C) of the GC for 1 min. The gas chromatograph was equipped with a ZB-5MS fused silica capillary column (30 m × 0.25 mm ID, df = 0.25 μm) from Phenomenex (Torrance, CA, USA). Hydrogen was used as carrier-gas with a constant flow rate of 3.1 ml/min. The temperature of the GC oven was raised from an initial 30 °C for 1.5 min, to 150 °C with 7.5 °C/min, followed by 10 °C/min to 250 °C and a final isothermal hold at 250 °C for 2 min. Electron ionization mass spectra were recorded at 70 eV with ion source and transfer line temperature of 230 and 250 °C, respectively. Only compounds with an abundance >1% across samples were further considered. Compound were identified based on their mass spectra, comparative database searches using the “Flavors and fragrances of natural and synthetic compounds 2” (Mondello, 2011) and Wiley2009/NIST2011 databases and if possible by comparison to authentic standards (see also Results). The absolute stereochemical configurations were not determined.

Statistics

As recently suggested by Junker (2018) we used a biosynthetically informed distance matrix (dA, B) to evaluate our data. Biosynthetically informed distances (dA, B) are similar to generalized unique-fraction-metric distances (UniFrac) used in microbial ecology (Chen et al., 2012; Lozupone & Knight, 2005). Different from the semi-metric Bray–Curtis similarity (Bray & Curtis, 1957) that only incorporates information on the composition of a community, UniFrac also considers the relative relatedness of the different community members by weighting their reciprocal phylogenetic distances (Chen et al., 2012; Lozupone & Knight, 2005). Yet, instead of including the bacterial phylogeny to community abundance data, the approach suggested by Junker (2018) incorporates the enzymatic origin or substance classes (e.g., aromatic, fatty acid, etc.) of compounds that co-occur in a complex blend with a specific relative composition. This method corrects for the problem of biochemical relatedness. It thus partly solves the problem that compounds of a compositional chemical dataset are dependent due to shared biosynthetic pathways. Biosynthetically informed distances (dA, B) require detailed knowledge about the enzymes involved. Often times, however, such information are not available, because biosynthetic pathways of many compounds are still unresolved (Morgan, 2010). Thus, Junker (2018) also developed an alternative approach which allows to incorporate chemical substance classes or other information such as compound chain length. Since no enzymatic data on the mandibular gland compounds of army ants are available, we used the alternative approach and used compound chain length as an additional information to calculate the biosynthetically informed distance matrix. We used chain length because it more accurately reflects the relationships between the studied compounds than compound classes. This is because ketones and alcohols of the same chain length are directly linked by a simple redox reaction. As suggested by Junker (2018), we merged this distance matrix with the conventional Bray–Curtis matrix using the default settings (see Junker, 2018 for a detailed outline, the R script, and R functions). We used the dA, B matrix to test for multivariate compositional differences with a permutational multivariate analysis of variance (PERMANOVA), to test for homogeneity of multivariate dispersions (PERMDISP), and to construct a non-metric multidimensional scaling (NMDS) ordination plot. Vectors in the ordination space, which represent compounds significantly contributing to data point separation, were fitted onto the NMDS plot as arrows, using the envfit()-function in “vegan” (Oksanen et al., 2007). Significance of fitted vectors was assessed using permutations (n = 10,000) and goodness of fit statistics (see Oksanen et al., 2007 and data supplement for details).

A well resolved phylogeny of Eciton was recently published (Winston, Kronauer & Moreau, 2017), of which we extracted the species divergence times. We used divergence times and dA, B values to investigate whether mandibular gland chemistry reflects the species’ phylogenetic history. For this, we constructed two cluster dendrograms of the divergence time and the merged biosynthetically informed distance dA, B using the tanglegram function as implemented in the R package “dendextend.” The tanglegram structure was improved to maximum overlap of both dendrograms by rotating the trees’ nodes without changing their topology. Additionally, we performed Mantel tests to analyze whether divergence time and chemical dissimilarity matrices are correlated with each other. As chemical dissimilarity matrices we used: (i) the merged biosynthetically informed distance dA, B, (ii) a Bray–Curtis dissimilarity matrix based on the mean relative proportion of compounds with a different chain length (categories: C6–C11, C13), and (iii) a Bray–Curtis dissimilarity matrix based on the mean relative proportion of different compound classes (i.e., alcohols, esters, ketones, indole). As we only correlated very small matrices (seven species) with each other, the statistical power was comparatively low. All analyses were performed with R 3.3.2 (R Core Team, 2017), using the R packages “ade4” (Dray & Dufour, 2007), “cluster” (Maechler et al., 2012), “dendextend” (Galili, 2015), “GUniFrac” (Chen, 2012) and “vegan” (Oksanen et al., 2007).

Results and Discussion

Compared to the study of Keegans et al. (1993), our analyses revealed a more complex chemical blend of mandibular gland VOCs in Eciton ants, which might be attributed to the development of more enhanced chemical analytical techniques. We detected and identified ten different VOCs from the mandibular glands of Eciton army ants and two additional ones from N. esenbeckii (Table 1). The main compounds were characterized by a prominent m/z = 86 or m/z = 59 fragment (in ketones or alcohols, respectively). The m/z = 86 indicated either 4-ketones or 3-ketones with a methyl group in position 2 or 4 arising from a McLafferty rearrangement of the carbon-oxygen double bond. This ion, the lack of a prominent m/z = 71, together with the base ion at m/z = 57 arising from the α-cleavage relative to the carbonyl C, indicated that the compounds are 4-methyl-3-ketones (5, 7, 9; see Table 1 for compound IDs). The identities of 4,6-dimethyl-3-ketones were evaluated based on mass spectrometric data provided by Fales et al. (1980), Bestmann et al. (1988) and Bergmann et al. (2001). Consequently, we assigned the detected alcohols based on their M+−1 ion to an affiliated ketone that represented the oxidized form of a respective alcohol (see also Do Nascimento et al., 1997). Similarly, we identified the 2-ketone (prominent ion at m/z = 58 again arising from a McLafferty rearrangement) and 2-alcohol (base ion at m/z = 45 representing a CH3-CH-OH moiety) as heptan-2-one (3) and heptan-2-ol (4), which are known glandular compounds of ants (see Blum et al., 1982; Scheffrahn et al., 1984). The identity of 4-methylheptan-3-one (5), 4-methylheptan-3-ol (6), 2-methylpentan-1-ol (1), and 3-methyl-1H-indole (11; common name: skatole) was additionally confirmed by authentic standards which were kindly provided by Stefan Schulz (TU Braunschweig) or purchased from Sigma-Aldrich (St. Louis, MO, USA). The ester 3-methylbutyl octanoate (12) was identified based on mass spectrometric data using the procedure described in Brückner et al. (2015). Most of the detected ketones (3, 5, 7, 9) and alcohols (1, 2, 4, 6, 8, 10) are common mandibular gland compounds of ants (Attygalle & Morgan, 1984). For instance, 4-methylheptan-3-one (5) and 4-methylheptan-3-ol (6), which were found in all species studied here, were described as alarm pheromones of several ant species including E. burchellii (Keegans et al., 1993; Hölldobler & Wilson, 1990). The same two compounds are also described as defensive compounds in several arthropods, for example, in opilionids (Meinwald et al., 1971; Raspotnig, 2012).

Table 1 Mass spectrometric data of volatiles organic compounds collected from the mandibular glands of army ants from the genus Eciton as well as N. esenbeckii.

Compound ID	Mass spectrometric fragmentation m/z (relative intensity %)	Identified as	
1	101 (M+−1; < 1), 84 (6), 71 (19), 70 (18), 69 (18), 57 (7), 55 (29), 43 (100), 41 (35)	2-methylpentan-1-ol	
2	115 (M+−1; < 1), 98 (2), 69 (19), 59 (100), 58 (24), 57 (26), 45 (35), 41 (38)	4-methylhexan-3-ol	
3	114 (M+; 5), 99 (3), 85 (3), 71 (15), 58 (61), 55 (7), 43 (100), 41 (13)	heptan-2-one	
4	115 (M+−1; 1), 101 (4), 98 (8), 83 (9), 70 (9), 55 (25), 45 (100), 41 (13)	heptan-2-ol	
5	128 (M+; 1), 99 (6), 86 (43), 71 (63), 57 (100), 55 (13), 43 (72), 41 (19)	4-methylheptan-3-one	
6	129 (M+−1; < 1), 112 (1), 101 (9), 83 (17), 70 (8), 59 (100), 55 (24) 43 (19), 41 (24)	4-methylheptan-3-ol	
7	156 (M+; < 1), 127 (1), 99 (5), 86 (44), 69 (3), 57 (100), 55 (6), 43 (11), 41 (14)	4,6-dimethyloctan-3-one	
8	157 (M+−1; < 1), 140 (4), 129 (8), 111 (11), 98 (7), 85 (3), 83 (4), 69 (40), 59 (100), 57 (35), 55 (22), 43 (20), 41 (28)	4,6-dimethyloctan-3-ol	
9	170 (M+; 1), 141 (1), 127 (1), 113 (3), 99 (6), 86 (77), 71 (43), 69 (5), 57 (100), 55 (13), 43 (51), 41 (28)	4,6-dimethylnonan-3-one	
10	171 (M+−1; < 1), 154 (1), 143 (7), 125 (3), 112 (7), 99 (1), 97 (2), 85 (10), 83 (17), 69 (35), 59 (100), 58 (28), 57 (33), 55 (25), 43 (40), 41 (26)	4,6-dimethylnonan-3-ol	
11	131 (M++1; 66), 130 (M+; 100), 103 (10), 77 (15), 51 (7)	3-methyl-1H-indole	
12	214 (M+; < 1), 171 (1), 145 (7), 144 (4), 128 (5), 127 (51), 115 (3), 101 (5), 71 (32), 70 (100), 57 (46), 55 (36), 43 (43), 41 (27)	3-methylbutyl octanoate	

Like in fungus-growing ants, army ant species had distinct mandibular gland chemical profiles (PERMANOVA: nperm = 10,000, n = 33, pseudoF = 55.5, r2 = 0.93, p < 0.001), forming idiosyncratic species clusters in the NMDS ordination plot (Fig. 1B; 2D-stress = 0.06; Table 2). Yet, some species exhibited more variable alarm pheromone profiles than others (PERMDISP: F6,27 = 5.2, p = 0.001). For instance, profile variance was higher in E. burchellii and E. hamatum compared to E. dulcium and E. lucanoides (Fig. 1B; Table 2). Interspecific differences in mandibular gland VOCs were not strictly determined by army ant phylogeny (Fig. 2). Only for E. burchellii and E. hamatum, we found that the closest relatives also possessed the closest similarity in mandibular gland VOCs (Fig. 2), with some E. burchellii headspace samples falling within the E. hamatum cluster (Supplemental Information S1). Interestingly, when tested against four allospecific army ant species, Torgerson & Akre (1970) observed that only E. burchellii crushed heads elicited a typical alarm response in E. hamatum workers. For all other Eciton species, the topologies of the two dendrograms, that is, species divergence times and biosynthetically informed distances, were incongruent (Fig. 2) and not correlated (Mantel test on Spearman’s rank correlation: ρs = −0.21, p = 0.78). Further chemical characters like the chain lengths or compound classes of chemicals found in the mandibular gland were also not correlated to the divergence times (Mantel tests on Spearman’s rank correlation: compound chain length: ρs = −0.24, p = 0.80; compound classes: ρs = −0.29, p = 0.87). The incongruent dendrogram topologies together with non-significant Mantel tests provide evidence that alarm pheromone blends are not determined by phylogeny. Pheromone evolution in Eciton army ants seems to follow a “saltational mode of evolution” expressed by a pattern of phylogenetic overdispersion (Baker, 2002; Roelofs et al., 2002; Symonds & Elgar, 2008), rather than evolving via small gradual changes in pheromone composition which would most likely result in phylogenetic clustering.

Table 2 Proportions (mean ± SE) of volatile mandibular gland secretions of different army ant species.

Compound ID	N. esenbeckii	E. burchellii	E. dulcium	E. hamatum	E. lucanoides	E. mexicanum	E. vagans	
1	–	–	–	22.7 ± 2.1	–	–	–	
2	–	2.6 ± 0.4	–	–	–	–	14 ± 0.9	
3	60.1 ± 1.5	–	–	–	–	–	–	
4	8.8 ± 0.5	–	–	–	–	–	–	
5	22.6 ± 0.3	29.2 ± 6.2	0.9 ± 0.2	26.1 ± 6	4.7 ± 0.6	1.8 ± 0.5	8.7 ± 0.5	
6	8.5 ± 1.3	42.1 ± 6.9	4 ± 0.6	20.2 ± 5.9	21.8 ± 0.8	5.6 ± 0.9	11.4 ± 0.7	
7	–	5 ± 1.4	16.7 ± 1.4	1.1 ± 0.4	20.1 ± 0.6	13.4 ± 1.7	37.5 ± 2.5	
8	–	1.8 ± 0.5	2.7 ± 0.3	1.3 ± 0.3	5.7 ± 0.2	10.8 ± 2.5	5.6 ± 0.4	
9	–	13.6 ± 3.3	68.9 ± 0.5	10.6 ± 2.9	44.2 ± 1.5	33.1 ± 4.1	21.7 ± 3.5	
10	–	5.6 ± 2.0	6.8 ± 0.7	17.9 ± 3.5	3.5 ± 0.3	21.8 ± 4.2	1.3 ± 0.2	
11	–	–	–	–	–	5.4 ± 1.2	–	
12	–	–	–	–	–	8.2 ± 4.3	–	
Sample size	5	5	5	5	5	4	5	
Notes:

Subspecies names are listed in “Materials and Methods.” Mean proportions and standard errors are based on the uncorrected integrated peak areas of the total ion chromatograms obtained via SPME-GC/MS (see Haberer et al., 2017). Compound IDs correspond to Table 1.

Figure 2 A chemo-evolutionary scenario of Eciton army ants.

Dendrograms (based on unweighted averages) of species divergence times (extracted from Winston, Kronauer & Moreau, 2017) and chemical distances of mandibular gland secretions. Colors correspond to Fig. 1.

The saltational mode of pheromone evolution leads to distinct pheromone blends among closest relatives, and thus possibly reduces or prevents interspecific responses to pheromones (e.g., bark beetles: Symonds & Elgar, 2004; ermine moths: Löfstedt, Herrebout & Menken, 1991; ants: Torgerson & Akre 1970; van Wilgenburg, Symonds & Elgar, 2011; Menzel, Schmitt & Blaimer, 2017a). This pattern was now detected for alarm pheromone blends in two of the most prominent ant groups of tropical rainforests, that is, fungus-growing ants (Norman et al., 2017) and army ants. It remains to be confirmed whether this is a universal pattern in communities of closely related ants. For social insect alarm pheromones in general, Leonhardt et al. (2016) suggested that natural selection acts on pheromone diversification and maintenance and thus ecological rather than phylogenetic effects are expected to shape alarm pheromone blends in co-occurring ant species (see also Menzel, Blaimer & Schmidt, 2017b). The ultimate mechanism of alarm pheromone diversification in ants (e.g., selection against interspecific cross-activity) remains unknown and calls for further investigation.

Conclusion

The exact nature of alarm pheromone communication in army ants and the underlying mechanism of pheromone diversification are still poorly understood. Conclusive bioassays using more complex blends—as described here—and tests with more realistic pheromone concentrations are needed to better understand which chemical compounds of the mandibular gland bouquet are in fact relevant in army ant alarm communication. By providing details about the glands’ chemical blends, the present study provides a valuable resource for such future studies.

Supplemental Information

Supplemental Information 1 Additional cluster analysis of all headspace samples.

Individual based cluster analysis (UPGMA on merged dA,B). Abbreviations: Eciton burchellii foreli Mayr 1886 (= Eb), E. dulcium crassinode Borgmeier 1955 (= Ed), E. hamatum Fabricius 1781 (= Eh), E. lucanoides conquistador Weber 1949 (= El), E. mexicanum s. str. Roger 1863 (= Em), E. vagans angustatum Roger 1863 (= Ev) and Nomamyrmex esenbeckii wilsoni Santschi 1920 (= No).

Click here for additional data file.

Supplemental Information 2 Datasets (gas chromatographic TIC data; substance class data; divergence times and R code) used in this study.

Click here for additional data file.

We thank Bryan Ospina for his assistance in the field, Adrian Pinto, Carlos de la Rosa, Bernal Matarrita Carranza, and the entire staff of La Selva Biological Station for their generous support. We are also grateful to Günther Raspotnig for his advice on compound identification, Stefan Schulz for providing authentic standards, as well as two anonymous reviewers for valuable comments.

Additional Information and Declarations

Competing Interests

Author Contributions

Field Study Permissions

Data Availability

The authors declare that they have no competing interests.

Adrian Brückner conceived and designed the experiments, performed the experiments, analyzed the data, contributed reagents/materials/analysis tools, prepared figures and/or tables, authored or reviewed drafts of the paper, approved the final draft.

Philipp O. Hoenle performed the experiments, contributed reagents/materials/analysis tools, approved the final draft.

Christoph von Beeren conceived and designed the experiments, performed the experiments, analyzed the data, contributed reagents/materials/analysis tools, authored or reviewed drafts of the paper, approved the final draft.

The following information was supplied relating to field study approvals (i.e., approving body and any reference numbers):

Field research and export permits were issued and approved by the Ministry of the Environment, Energy and Technology of the Republic of Costa Rica (MINAET; permit numbers: 192-2012-SINAC and R-009-2014-OT-CONAGEBIO).

The following information was supplied regarding data availability:

The raw data and references to code are provided in the Supplemental files.

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
