# Peer review of "Comparative chemical analysis of army ant mandibular gland volatiles (Formicidae: Dorylinae)"

_PeerJ, doi:10.7717/peerj.5319_

## Round 0.1 · original submission · Minor Revisions

Although both reviewers were very positive, they identified several minor issues that have to be addressed.

Reviewer 1 ·

Basic reporting

See comment about literature reference under point 3, validity of the findings, perhaps to be included in the introduction as well.

Experimental design

Materials and methods are not clear on the point of sample size and origin, and thus do not provide sufficient detail and information to replicate:

Despite stating the number of colonies per species from which majors were collected, the number of workers per species included for analyses is not mentioned at this point. The information is eventually provided in Table 2, but still is ambiguous. There is no indication as to how many of the majors that were included in the analysis came from the same and/or different colonies.
As it stands, sample selection seems arbitrary: on which basis were the 5 workers selected from the different colonies for species in which 2 or 4 colonies were sampled?
Could there be differences between colonies (not expected, but it might be)?
Also, it is not clear whether the "colonies" really are different colonies (how was this ascertained?) as samples are reported to have been collected over 2 months and in a restricted area.

Please provide the respective information in the Materials and methods chapter.

Validity of the findings

You do not cite Torgerson and Akre (1970, Interspecific responses to trail and alarm pheromones by New World army ants), who report on interspecific reactions to alarm pheromones in several of the species that are analysed in the present study. Despite the fact that they do not report the chemical composition of mandibular glands, I believe at least the discussion would benefit from including the findings of that study, especially in regard to shared compounds as reported in the present manuscript.

Additional comments

lines 8-9: "little is known about their chemical communication" - this manuscript does not provide any further insight in that direction, either, and thus is somewhat misleading. I would rephrase.

line 130: the reference for R is not correct, please change to "R Core Team, 2017"

line 175: why is it remarkable that a communication signal does not follow phylogeny? If there are reports of chemical communication signals of ants that actually are determined by phylogeny, please provide the relevant references.

Reviewer 2 ·

Basic reporting

The language is good and understandable. But I miss some biological background. As it is now, the study seems to have a very narrow focus – although the evolution of chemical bouquets is a field of high interest to people from many biological fields! The authors should put their study into a broader context and e.g. write more about chemical evolution, and phylogenetic signal, in other taxa and other substance classes. There are some studies on phylogenetic constraints in beetle pheromones and in ant cuticular hydrocarbons, for example; probably there are many studies on the evolution of floral scents or other plant secondary metabolites.
Minor comments.
line 31: remove semicolon after “sensu”
lines 31-32: the two sentences are not really connected, make sure their content is linked better.
in line 33, you mention “ecological selection”, in line 179 you write “natural selection”. I guess the same is meant in both cases; you should use a consistent term to avoid confusion.
line 76: no differences… chemical differences or differences in the behaviour they evoke?

Experimental design

The research question (lines 48-52) is well-defined and relevant. However, it seems narrow as it is presented. The authors should try to make it more general and put it into the context of chemical evolution.
The chemical methods are clearly described and understandable, and the same is largely true for the statistical analysis. However, I suggest that the authors provide the detailed classification for each compound that entered the analysis. This could be easily done with an additional column in Table 1. Furthermore, the authors should describe how the determined which substances significantly contributed to data separation ( in Fig. 1B), and how the arrows were calculated. In fig. 1B, the authors may consider adding the species initials (“Ne”,”Eb”,…) to each dot so that the figure can be interpreted from a black-and-white copy as well. Did the relative abundances of the compounds also enter the analysis?
line 80: Spelco or Supelco?
line 113: what do you mean by “weighted in”?
line 128: I guess you did not randomly rotate the nodes, but had a goal… so I would rather write “… by rotating certain tree nodes…”
line 130: the reference for R is incorrect.

Validity of the findings

The data are robust and statistically sound. The authors write only little about the implications of their findings, this should be expanded. For example, they suggest that ecological effects may shape alarm pheromone blends (lines 180-182) – here they could become more concrete and suggest more precisely which effects may promote pheromone diversification. Also, put the results in the context of other studies on the evolution of chemical blends.
I suggest that the authors re-run the Mantel test with other types of classification (e.g. only chain length, only substance class) to see whether there is phylogenetic signal in one of the chemical traits. It should also be mentioned that the statistical power is low – given an n of 7 species, the Mantel test is less likely to become significant anyway.

Additional comments

no comment

---

## Round 0.2 · accepted · Accept

I believe that you properly addressed all of the issues raised by the reviewers.

#